# GROUNDATTACK: MITIGATING EASY-OPTIONS BIAS FOR VISUAL QUESTION ANSWERING

## ABSTRACT

In this early study, we observe an **Easy-Options Bias** (EOB) issue in several multiple-choice Visual Question Answering (VQA) benchmarks, including MM-Star, RealWorldQA, SEED-Bench, NExT-QA, STAR benchmark, and Video-MME. This bias allows vision-language models (VLMs) to select the correct answer using only the vision ($V$) and options ($O$) as inputs, without the need for the question ($Q$). Through grounding experiments, we attribute the bias to an imbalance in visual relevance: the correct answer typically aligns more closely with the visual contents than the negative options in feature space, creating a shortcut for VLMs to infer the answer via simply vision-option similarity matching. To mitigate this, we introduce **GroundAttack**, an agentical method that automatically generates hard negative options as visually plausible as the correct answer. We apply it to the NExT-QA and MMStar datasets, creating new EOB-free annotations. On these EOB-free annotations, current VLMs approach random accuracies under ($V+O$) settings, and drop to non-saturated accuracies under ($V+Q+O$) settings, providing a more realistic evaluation of VLMs' QA ability.

## 1 INTRODUCTION

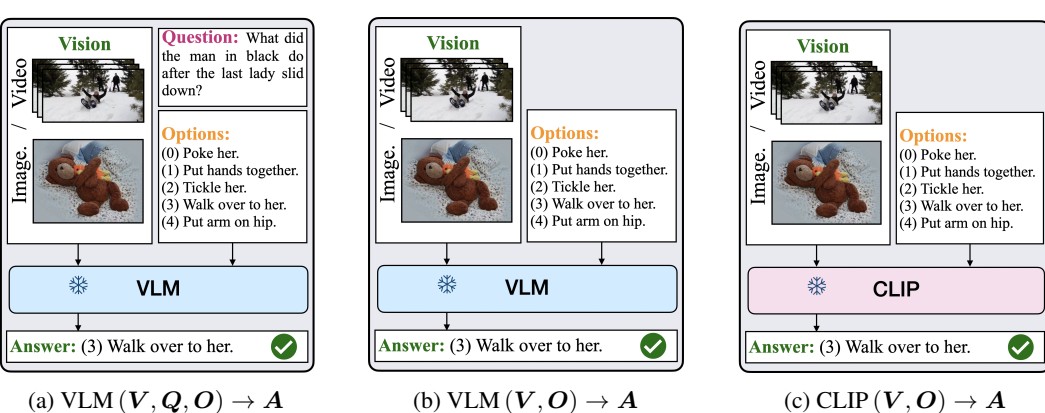

(a) VLM $(V, Q, O) \to A$          (b) VLM $(V, O) \to A$          (c) CLIP $(V, O) \to A$

Figure 1: **Easy-Options Bias** lets a VLM pick the correct answer without seeing the question. Here, $V, Q, O, A$ denote the vision input, question, options, and the correct answer.

Visual Question Answering (VQA) is a core benchmark in multimodal research, designed to test a model's ability to jointly reason over visual and linguistic inputs (Yu et al., 2019; Xiao et al., 2021; Wu et al., 2021; Yue et al., 2024; Chen et al., 2024b). In particular, multiple-choice VQA tasks require a model to select the correct answer from a set of options given an image and a natural language question. Such tasks are widely used to evaluate vision-language models (VLMs), under the assumption that correct performance necessitates understanding and integrating both the visual content and the question semantics.

Over the past few years, VLMs have made remarkable progress across VQA benchmarks, often surpassing human-level performance. These gains have been enabled by advances in large-scale

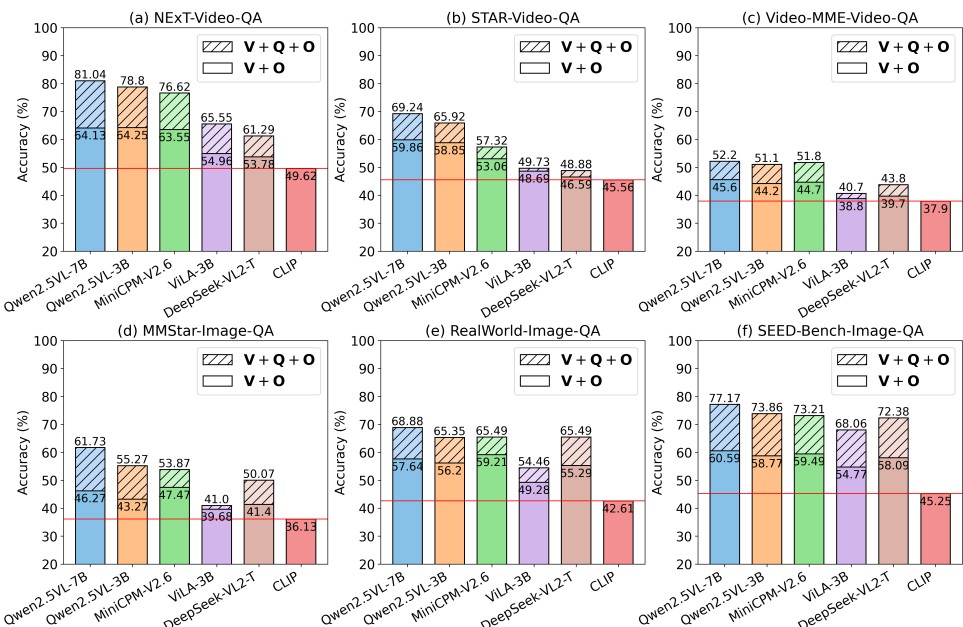

Figure 2: **Easy-Options Bias across six VQA benchmarks and four VLM series.** Across all datasets and models, VLMs with "vision+options" inputs achieve an mean accuracy of 52.27%, just 9.07% lower than the 61.34% mean accuracy with "vision+question+options" inputs. When use CLIP to select the most visually similar options ("vision+options"), the mean accuracy reaches 42.82%. This shows that negative options are less groundable than the correct answer in these VQA benchmarks, creating shortcuts that VLMs can exploit.

pretraining, attention-based architectures, and integration of vision and language modalities via contrastive and generative learning. However, there is growing concern that such improvements may not reflect genuine multimodal reasoning. Instead, models may be exploiting dataset biases (Rawal et al., 2024), superficial correlations, or artifacts in the benchmark design, echoing earlier concerns in natural language processing, where models often succeed through spurious cues rather than deep understanding (Geirhos et al., 2020; Chao et al., 2017; Yang et al., 2020; Manjunatha et al., 2019; Cadene et al., 2019; Clark et al., 2019; Zhong et al., 2022).

Before the VLM/LLMs era, **shortcut learning** (Dancette et al., 2021) posed a critical challenge for Visual Question Answering (VQA). Shortcut learning happens when a model relies on shallow patterns in the <vision, question> inputs instead of truly understanding or reasoning about the contents. For example, on some VQA benchmarks, "what colour..." questions often get the answers "white". These biases emerge because the answer distributions for colour-relevant questions are long-tailed, with "white" appearing most frequently. During training, VQA models unconsciously learn these statistics. Since most VQA datasets split the training/testing sets in an identical distribution manner (IID), models can exploit these shortcuts to achieve spurious high accuracy on the testing set.

Several well-known biases in VQA benchmarks that lead to shortcut learning include language bias (Goyal et al., 2017), texture bias (Geirhos et al., 2018), and type bias (Agrawal et al., 2018). Their key characteristic is that the model predicts the most frequent associated answer whenever a particular visual or linguistic cue appears. As a result, VQA models often reach performance saturation on standard benchmarks but generalize poorly to out-of-domain inputs. To reduce shortcut learning, researchers have created de-biased VQA benchmarks. They rebalance the testing set so it no longer mirrors the training set (OOD), forcing models to go beyond shallow patterns and combine vision and language. Representative works include VQA-CP (Agrawal et al., 2018), VQA-CE (Dancette et al., 2021), VQA-VS (Si et al., 2022), and GQA-AUG (Reich & Schultz, 2024). By breaking the training–testing correlations, these benchmarks compel models to learn beyond superficial shortcuts and focus on genuine understanding and reasoning.

In the VLM/LLM era, VLMs benefit from pre-training on web data, and show strong performances on existing VQA benchmarks under *zero-shot* conditions. Unlike predecessor VQA models, which were trained on publicly available training data, VLMs are trained behind the scenes on large-scale, weakly labeled web data by industrial companies. That means we can't guarantee their training and testing sets meet the OOD standards. Besides, VLMs store more prior knowledge than predecessor VQA models and learns human-like answering skills; they can exploit even subtler shortcuts than before. For example, Chen et al. (Chen et al., 2024a) show that VLMs suffer from a language bias: they can predict the correct answer using only the question, without looking at the image. They call this a "*Lack of Visual Dependency*" issue in VQA benchmarks. For example, given questions like "Which model achieves the best ImageNet accuracy?", a VLM can answer "SoftMoE" correctly without looking at the image input. This language-only shortcut occurs because VLMs memorize prior knowledge from web-scale data, enabling them to make correct guesses.

In this paper, we observe a new **Easy-Options Bias** (EOB) in multiple-choice VQA benchmarks when testing VLMs (see Figures 1a–1b). EOB happens when VLMs consider the negative options so irrelevant to vision inputs that they no longer require the question. In other words, given only "*vision+options*", a VLM can pick the correct answer just as well as if it saw the "*vision+question+options*". To verify that this bias holds across tasks and models, we evaluate on six VQA benchmarks: three video datasets (Xiao et al., 2021; Wu et al., 2021; Fu et al., 2024) and three image datasets (Chen et al., 2024a; xAI, 2024; Li et al., 2024). We tested four types of SOTA VLMs (Bai et al., 2025; Yao et al., 2024; Lin et al., 2024; Wu et al., 2024). When given only the vision inputs and the answer choices, VLMs still score an average of 52.27% across all datasets and models (i.e., VLM $(V, O)$). That's surprisingly high, just 9.07% below the 61.34% average when VLMs also see the question (i.e., VLM $(V, Q, O)$) (see Figure 2).

This phenomenon exposes a critical flaw in current VQA benchmarks: if vision–language models can reliably select the correct answer without reading the question, then benchmark accuracy is no longer a valid proxy for multimodal reasoning. We hypothesize that EOB may arise from four sources: (1) visually biased answer sets, where the correct option aligns more strongly with the image than the distractors; (2) question redundancy, where the question adds little beyond what is already implied by the image and options; (3) shortcut learning from spurious, dataset-specific correlations; and (4) language priors that favor statistically plausible choices irrespective of context.

To probe the cause of EOB, we conduct a grounding test with the CLIP model (Radford et al., 2021; Zhai et al., 2023). We compute the similarity between the visual embedding of the image (or sampled video frames) and the text embeddings of the answer options, without conditioning on the question. Across all evaluated benchmarks, a consistent pattern emerges: *the correct answer attains higher image–text similarity than the distractors* (see Figure 1c and Figure 2). This reveals a pronounced visual relevance imbalance: the correct answer is not only semantically appropriate but also more strongly grounded in the visual content than the negatives, so that CLIP alone can often pick the correct answer via similarity matching.

Addressing EOB is challenging given the inherent limitations of existing datasets. We therefore propose **GroundAttack**, a practical mitigation method that generates visually and semantically plausible hard negatives to rebalance option relevance. Experiments show that GroundAttack substantially reduces the impact of EOB across benchmarks, yielding more robust VQA evaluation.

## 2 GROUNDATTACK: CREATING GROUNDABLE ADVERSARIAL NEGATIVE OPTIONS

**Definition 1:** *Given a visual input* $V$, *Question* $Q$, *options set* $O$ *as potential answers and the correct answer* $A \in O$, *if a model* $\pi(V, O)$ *predicts or select the correct option answer* $A$, *without utilizing the Question* $Q$ *as an input to* $\pi(\cdot)$, *then that question Q suffer from* **Easy-Options Bias under** $\pi$.

**Definition 2:** *Given* $V, Q, O, A$ *tuple, if any one of the model* $\pi$ *from a set of models* $\Pi$ $(\pi \in \Pi)$ *predicts or select the correct option answer* $A$, *without utilizing the Question* $Q$ *as an input to it, then that question* $Q$ *suffer from* **Easy-Options Bias**.

**Definition 3:** *Given* $V, Q, O, A$ *tuple, if every model* $\pi$ *in a given set of models* $\Pi$ *selects the correct answer option* $A$, *without utilizing the Question* $Q$ *as an input to them, then that question* $Q$ *suffer from* **Total Easy-Options Bias**.

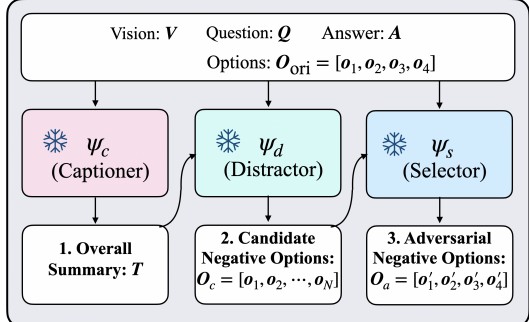

Figure 3: **GroundAttack** generates adversarial negative options that are more confusing, diverse, and visually groundable than original negatives. It mitigates Easy-Options Bias in VQA benchmarks through three components: (1) the Captioner ($\psi_c$), which converts visual content into detailed descriptions; (2) the Distractor ($\psi_d$), which produces plausible, groundable negative candidates; and (3) the Selector ($\psi_s$), which identifies the most adversarial negatives.

**Lemma 1:** *If we assume all models in $\Pi$ predicts randomly without the question as input, the expected proportion of questions suffer from **Easy-Options Bias** for a given benchmark is given by $1 - [(1 - \frac{1}{|O|})^{|\Pi| - (\lambda - 1)}]$ where $\lambda$ accounts for the number of correlated models ($0 \leq \lambda \leq |\Pi|$).*

Table 1 presents the proportion of questions exhibiting Easy-Options Bias and Total Easy-Options Bias across a range of foundation models, including Qwen2.5VL-7B, Qwen2.5VL-3B, MiniCPM-V2.6, ViLA-3B, DeepSeek-VL2-Tiny. Here, "correlated models" refers to VLMs likely fine-tuned on overlapping training-set and therefore behaving similarly. If we assume the VLM behaviour is not random, then remarkably, over 80% of questions across all evaluated benchmarks are affected by Easy-Options Bias (Table 1). Even more striking is the substantial fraction of questions that exhibit Total Easy-Options Bias, particularly in SEED-Bench and NExT-QA, indicating that in many cases, models can reliably predict the correct answer without any access to the question. It is also worth noting that among all the benchmarks considered, MMStar exhibits the lowest incidence of Easy-Options Bias. However, the issue remains significant and cannot be overlooked, underscoring the need for more robust evaluation.

Motivated by the recent works in distractor generators such as (Çavuşoğlu et al., 2024; Chung et al., 2020), we introduce **GroundAttack** (see Figure 3), which addresses the Easy-Options Bias in existing VQA benchmarks. Our GroundAttack replaces only the negative options in each tuple, while preserving the original vision, question, and answer. It consists of three modules: *Captioner* $\psi_c$, which converts visual inputs into concise textual descriptions $\mathbf{T}$; *Distractor* $\psi_d$, which generates visually grounded candidate options $\mathbf{O_c}$; and *Selector* $\psi_s$, which mines adversarial hard negatives $\mathbf{O_a}$. We implement these modules as collaborating agents, making GroundAttack a pragmatic approach that minimizes human effort (Equations 1a-1c).

$$\psi_c(\boldsymbol{V}) \rightarrow \boldsymbol{T}, \quad (1a) \qquad \psi_s(\boldsymbol{V}, \boldsymbol{O}_{\text{ori}}, \boldsymbol{O}_c) \rightarrow \boldsymbol{O}_a, \quad (1c)$$
$$\psi_d(\boldsymbol{Q}, \boldsymbol{A}, \boldsymbol{T}) \rightarrow \boldsymbol{O}_c, \quad (1b)$$

Table 1: Ratio of questions that suffers from Easy-Options Bias and Total Easy-Options Bias under four types of SOTA VLMs (Qwen2.5VL-7B, Qwen2.5VL-3B, MiniCPM-V2.6, ViLA-3B, and DeepSeek-VL2-Tiny).

| Percentage of Biased Samples | NExT-QA | STAR-QA | MMStar | RealWorld | SEED-Bench |
|---|---|---|---|---|---|
| *Easy-Options Bias* | 87.04% | 85.24% | 78.86% | 93.72% | 80.23% |
| *Total Easy-Options Bias* | 30.07% | 20.56% | 13.60% | 17.64% | 35.38% |

**Captioner** $\psi_c$   Given a video or image input $\boldsymbol{V}$, model $\psi_c$ serves to convert it into a concise text description $\boldsymbol{T}$. Many off-the-shelf visual captioning models are available, such as BLIP (Li et al., 2022). These models are either trained on domain-specific video or image datasets, and fit only a small range of captioning tasks. We also want the $\psi_c$ to extract fine-grained information, such as object locations, actions, and spatial relations. We resort to large-scale pre-trained VLMs as our captioner. This choice removes the need to pick and customize separate captioning models, ensuring GroundAttack (GDA) is compatible with different video and image benchmarks.

We use prompt engineering to guide the VLM in captioning complex visuals into detailed descriptions $\boldsymbol{T}$. A simplified prompt is shown below; full prompts appear in the supplementary material.

```
You are an expert that extracts descriptive facts, detailed object
information (bounding boxes), actions, scene context, and spatial
relations from images or videos, ...:
```

**Distractor** $\psi_d$   Given the visual description $\boldsymbol{T}$, the question $\boldsymbol{Q}$, and the correct answer $\boldsymbol{A}$, the agent $\psi_d$ creates $N$ negative choices:$\boldsymbol{O}_c = [\boldsymbol{o}_1, \boldsymbol{o}_2, \ldots, \boldsymbol{o}_N]$. In past VQA benchmarks, researchers did this by hand: they either wrote rules by hand (Wu et al., 2021) or hired annotators to check wrong answer options (Xiao et al., 2021). This manual process was slow but needed, since each wrong choice must be clearly wrong yet still confusing.

We employ an LLM as an agent to automatically generate negative answer choices using simple prompts. This approach reduces human effort and mitigates inter-annotator inconsistency, since the same LLM produces all negatives. Below we include an abbreviated version of our prompt; the full prompt is provided in the supplementary material.

```
You are an expert at generating challenging negative distractors
for image-based question answering.
Given an image description, a question, and its correct answer,
generate 128 clearly and definitively incorrect answer options...
{Question}, {Correct Answer}, {Description}, Negative Options:
```

**Selector** $\psi_s$   Given the candidate negatives $\boldsymbol{O}_c$ and the original negatives $\boldsymbol{O}_{\mathrm{ori}}$, the agent $\psi_s$ chooses a subset $\boldsymbol{O}_a$ that both confuses the model and keeps enough variety among the options. We implement three simple strategies: *random sampling*, *CLIP selector*, and *clustering + CLIP*. We experimentally verify that "clustering + CLIP" introduces enough confusion of negative options while maximizing options' diversity. Below, describe the three $\psi_s$.

*Random sampling* selects a fixed number of negative options uniformly from $\boldsymbol{O}_c$. This simple baseline is common in prior work on negative-option generation for VQA benchmarks, but it does not directly address Easy-Options Bias; it merely relies on the proposal agent $\psi_d$ to produce sufficiently confusing distractors.

*CLIP selector* ranks candidate options by their visual similarity to the input $\boldsymbol{V}$ and selects hard negatives accordingly. First, it encodes all $N$ candidate options $\boldsymbol{O}_c$ into text features $\{\boldsymbol{f}_c^{(i)}\}_{i=1}^N$ with the CLIP text encoder (Eq. 2a). It then encodes the visual input with the CLIP vision encoder; for video, it applies temporal average pooling $\mathcal{F}_{\mathrm{tavg}}$ to aggregate frame features into a single representation (Eq. 2b), whereas no temporal pooling is required for images. Next, it computes similarities between each $\boldsymbol{f}_c^{(i)}$ and the visual feature and ranks candidates by these scores. Finally, it selects the top-$m$ options with the highest similarity to $\boldsymbol{V}$ as hard negatives (Eq. 2c).

.

$$\boldsymbol{f}_c = \mathcal{F}_{\mathrm{CLIP\text{-}Text}}(\boldsymbol{O}_c)\,, \tag{2a}$$

$$\boldsymbol{f}_v = \mathcal{F}_{\mathrm{tavg}}\big(\mathcal{F}_{\mathrm{CLIP\text{-}Vision}}(\boldsymbol{V})\big)\,, \tag{2b}$$

$$\boldsymbol{O}_a = \{\boldsymbol{o}_{i^*}\}_{i=1,2,\cdots,m}\,, \quad i^* = \arg\max_i\big[(\boldsymbol{f}_c\,\boldsymbol{f}_v^\top)_i\big]\,, \tag{2c}$$

$$\boldsymbol{f}_c \in \mathbb{R}^{N \times D}\,, \quad \boldsymbol{f}_v \in \mathbb{R}^{1 \times D}\,.$$

*Clustering + CLIP* selects adversarial negatives in two stages. (i) We apply $k$-means to cluster the candidate negatives into $m$ groups based on their text features. (ii) Within each group, we use CLIP to select the highest-similarity (top-1) option to the visual input $V$. The union of these selections yields $m$ groundable negatives, denoted $O_a$.

$$[f_{c1}, f_{c2}, \cdots, f_{cm}] = \text{K-Means}(f_c, m),$$ (3a)

$$O_a = \{o_{i^*}\}_{j=1,2,\cdots,m}, \quad i^* = \arg\max_1 [(f_{cj} f_v^\top)_i],$$ (3b)

We test different strategies for $\psi_s$ in §3 and select "Clustering+CLIP" as it ensures that adversarial negative options are confusing, representative, and groundable. To this end, we built the GroundAttack toolkit with three frozen foundation models (i.e., VLM, LLM and CLIP) in an agent-collaborating manner.

## 3 EXPERIMENTS

### 3.1 DATASET & SETTING.

**MMStar** (Chen et al., 2024a) is a mixed ImageQA benchmark constructed from six existing datasets that reduce language biases. It includes an evaluation-only set of 1,500 samples spanning six categories: coarse perception, fine-grained perception, instance reasoning, logical reasoning, science & technology, and mathematics. We apply GroundAttack to generate new adversarial negative options for MMStar, except for the science and mathematics categories, whose questions are largely textbook-style; in these cases, GroundAttack does not outperform expert-curated options. We will release the resulting GDA-Annotation to support future research.

**NExT-QA** (Xiao et al., 2021) is a widely used VideoQA benchmark comprising 5,440 videos and 34,132/4,996 QA samples in the training and validation sets. It covers causal, descriptive, and temporal question types. However, its negative options are randomly sampled from similar questions and manually verified, a strategy akin to $\psi_s$ = "random sampling", which VLMs can easily exploit with Easy-Options Bias. To address this, we use GroundAttack to generate adversarial negative options for NExT-QA and will release the GDA-annotation for future research.

**Settings** We use GLM-4.1V-9B (Hong et al., 2025) as the captioner ($\psi_c$), Google Gemma-3n-E48 (Team et al., 2025) as the distractor generator ($\psi_d$), and MetaCLIP-2 (Chuang et al., 2025) as the selector ($\psi_s$). We purposely choose latest LLM/VLM/CLIP agents that are different from those used by the evaluated VQA models (VLMs) to avoid leakage and prevent model-specific advantages. Using $\psi_d$, we generate $N = 128$ candidate answers and apply GroundAttack to respectively generates $m$=3/4 adversarial negatives for the MMStar or NExT-QA benchmarks, matching the original number of negatives. For VideoQA, we sample 8 frames per video. All experiments are run on a single NVIDIA A100 GPU (80 GB).

### 3.2 ANALYSIS.

**Comparisons of different negative options.** We compare negative options generated by the original method, random sampling, CLIP-selector, and GroundAttack in Tables 2 and 3. All experiments are evaluated under the $(V, Q, O)$ setting unless otherwise specified.

We observe that: (1) GroundAttack significantly decreases accuracies across all five VLMs compared to the original negative options, when Easy-Options Bias is mitigated on the MMStar benchmark. For example, Qwen2.5VL-7B drops from 61.73% to 52.00%, and DeepSeek-VL2-Tiny decreases from 50.07% to 36.80%. A similar observation appeared on the NExT-QA benchmark. (2) Using random sampling in the selector $\psi_s$ introduces minimal confusion on the MMStar benchmark (random: 62.73% *vs.* original: 61.73% with Qwen2.5VL-7B). This suggests that random sampling fails to meet the necessary criteria that negative options are confusing. (3) Both the CLIP-Selector and GroundAttack (Clustering+CLIP) effectively mitigate Easy-Options Bias on the two benchmarks. Compared to the CLIP-Selector, GroundAttack selects more diverse and representative negatives

and produces stronger adversarial options. (4) We further evaluate the original and GroundAttack-generated options under the $(\boldsymbol{V}, \boldsymbol{O})$ setting, where the question is omitted. In this setting, VLM accuracies drop to the 20%–30% range, close to random guessing, compared to 50%–40% in the standard $(\boldsymbol{V}, \boldsymbol{Q}, \boldsymbol{O})$ setting on the NExT-QA and MMStar benchmarks, respectively.

Table 2: Comparison of VLM performance with different negative option strategies on the MMStar benchmark. ($\downarrow$) indicates that lower values reflect more distracting (and thus better) negative options.

| Negative Options | VLM Qwen2.5VL-7B ($\downarrow$) | Qwen2.5VL-3B ($\downarrow$) | MiniCPM-V2.6 ($\downarrow$) | ViLA-3B ($\downarrow$) | DeepSeek-VL2-Tiny ($\downarrow$) |
|---|---|---|---|---|---|
| Original Chen et al. (2024a) | 61.73 | 55.27 | 53.87 | 41.00 | 50.07 |
| Random Negatives | 62.73 | 57.67 | 54.80 | 42.68 | 45.80 |
| CLIP-Selector | 52.47 | 48.47 | 46.80 | **30.13** | 41.60 |
| **GroundAttack** | **52.00** | **46.87** | **45.53** | 32.33 | **36.80** |
| Original (V,O) | 46.27 | 43.27 | 47.47 | 39.68 | 41.40 |
| **GroundAttack** (V,O) | **28.33** | **27.67** | **29.33** | **24.73** | **25.53** |

Table 3: Comparison of VLM performance with different negative option strategies on the NExT-QA benchmark. ($\downarrow$) indicates that lower values reflect more distracting (and thus better) negative options.

| Negative Options | VLM Qwen2.5VL-7B ($\downarrow$) | Qwen2.5VL-3B ($\downarrow$) | MiniCPM-V2.6 ($\downarrow$) | ViLA-3B ($\downarrow$) | DeepSeek-VL2-Tiny ($\downarrow$) |
|---|---|---|---|---|---|
| Original Xiao et al. (2021) | 81.04 | 78.80 | 76.62 | 65.55 | 61.29 |
| Random Negatives | 71.14 | 69.98 | 69.08 | 50.14 | 37.83 |
| CLIP-Selector | 54.96 | 53.02 | **52.18** | **31.59** | **21.06** |
| **GroundAttack** | **54.16** | **52.66** | 53.24 | 33.21 | 24.98 |
| Original (V,O) | 64.13 | 64.25 | 63.55 | 54.96 | 53.78 |
| **GroundAttack** (V,O) | **20.80** | **26.30** | **26.28** | **21.66** | **15.13** |

**Impacts of the number of GroundAttack negative options.** In Figure 4, we study the effect of varying the number of GroundAttack-generated negative options. We use Qwen2.5VL-7B as the evaluation model and report results under both $(\boldsymbol{V}, \boldsymbol{O})$ and $(\boldsymbol{V}, \boldsymbol{Q}, \boldsymbol{O})$ settings. Because the original NExT-QA and MMStar annotations contain only 3 or 4 negatives, we extend random sampling as an approximate surrogate to mimic the original annotation size for reference.

We observe that accuracy decreases as the number of GroundAttack-generated negatives increases in both settings, unsurprising, as more confusing distractors make it harder for VLMs to select the correct answer. Notably, under $(\boldsymbol{V}, \boldsymbol{Q}, \boldsymbol{O})$, performance with GroundAttack is consistently lower than with random sampling, indicating that GroundAttack produces more optimally confusing negatives. Moreover, under $(\boldsymbol{V}, \boldsymbol{O})$, the GroundAttack curve moves closer to chance performance than the random-sampling baseline, suggesting that Easy-Options Bias is mitigated by GroundAttack. We cap the number of options at 26, corresponding to labels A–Z.

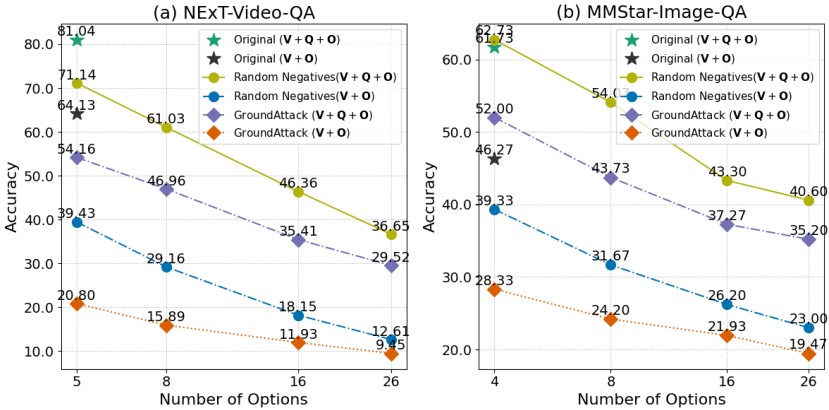

Figure 4: Impact of varying the number of GroundAttack-generated negative options on NExT-QA and MMStar benchmarks (Number of Options=1 Positive + $N$ Negatives)

In Table 4 we compute the Easy-Options Bias proportion after GroundAttack on both NExT-QA and MMStar datasets. Evidence suggests that our GroundAttack enforces to models behave randomly achieving both theoretical expected performance as per the *Lemma 1* for Easy-Options Bias.

Note that the theoretical expected proportion for random models for both datasets is around 48.80%∼59.04% for NExT-QA ($\lambda \in \{1,2\}, |\Pi| = 5, |O| = 5$), and 57.81%∼68.35% for MM-Star ($\lambda \in \{1,2\}, |\Pi| = 5, |O| = 4$).

### 3.3 VISUALIZATION

We present a visual comparison between the original negatives and adversarial negatives generated by GroundAttack, as shown below on the MMStar dataset. We observe that GDA options are more visually relevant to image contents than original annotations. (Figure 5).

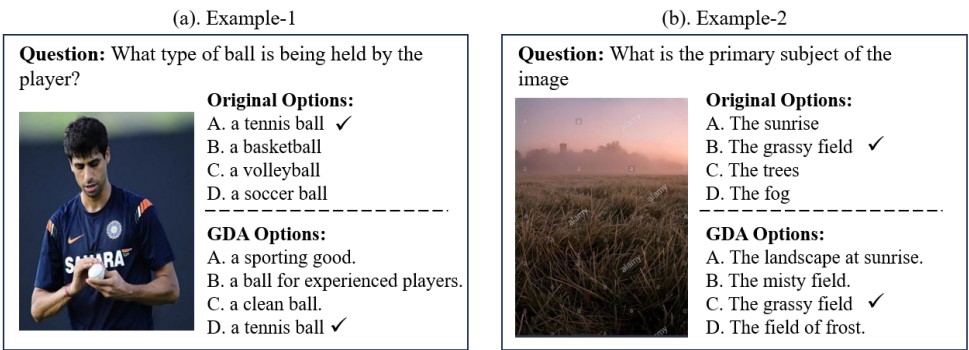

Figure 5: Visual comparison between original options and those generated by GroundAttack.

## 4 RELATED WORKS

Several studies have uncovered critical limitations in visual question answering (VQA) (Song et al., 2023; Qraitem et al., 2023; Agrawal et al., 2022), yet these findings have received limited attention in developing mainstream benchmarks. Sheng et al. (2021) demonstrated that VQA models fail dramatically when evaluated on datasets constructed using dynamic, human-adversarial approaches. Similarly, Li et al. (2021) introduced an adversarial benchmark to expose model vulnerabilities, while Zhao et al. (2023) examined the fragility of vision-language models (VLMs) in VQA settings. Shortcut learning in VQA, where models exploit superficial correlations across modalities, has also been addressed in several works (Dancette et al., 2021).

Zhang et al. (2023) identified a form of vision-answer bias in the NExT-QA dataset(Xiao et al., 2021), where the distribution of answers conditioned on visual inputs is skewed, leading to disproportionate vision-answer correlations. However, their analysis was confined to conventional VQA models and did not consider recent VLMs. In this work, we reveal an even more severe issue **Easy-Options Bias**, where models can often ignore the question entirely and still predict the correct answer based solely on the visual input and answer choices.

Wang et al. (2023) further identified two systematic dataset biases: (1) Unbalanced Matching, where the correct answer exhibits stronger alignment with the image and question than the distractors, and (2) Distractor Similarity, where incorrect answers are both dissimilar to the correct one and mutually

| (a) NExT-QA | | | | (b) MMStar | | |
|---|---|---|---|---|---|---|
| Percentage | *Easy-Options Bias* (↓) | *Total Easy-Options Bias* (↓) | | Percentage | *Easy-Options Bias* (↓) | *Total Easy-Options Bias* (↓) |
| Original Xiao et al. (2021) | 87.04% | 30.07% | | Original Xiao et al. (2021) | 78.86% | 13.60% |
| **GroundAttack** | 50.99% | **3.34%** | | **GroundAttack** | 59.33% | **3.40%** |

Table 4: By using GroundAttack, ratio of questions that suffer from Easy-Options Bias and total Easy-Options Bias under four types of SOTA VLMs. (↓) indicates that less samples suffer from Easy-Options Bias.

similar, thereby reducing discriminative challenge. A recent survey by Ma et al. (2024) provides a comprehensive overview of robustness in VQA.

In contrast to prior work, we focus on how state-of-the-art VLMs, including Qwen-VL-2.5 and DeepSeek-VL2, behave under modern evaluation settings such as the MMStar, SEED-Bench, and benchmarks. Our findings show the urgent need for more diagnostic evaluation protocols that account for vision-answer biases and question insensitivity in contemporary VQA tasks.

## 5 LIMITATIONS AND CONCLUSIONS

**Limitations**

Our findings are based on empirical observations and come with several limitations. First, while we analyzed the **EOB** phenomenon across a representative set of benchmarks and VLMs, our coverage is not exhaustive due to computational and resource constraints. Consequently, the generalizability of our conclusions may be limited.

Second, although our proposed strategy shows promise in reducing **EOB** on the datasets and models tested, we do not claim it universally addresses the issue. Its effectiveness may vary depending on dataset properties, model architectures, and training regimes.

Third, our analysis emphasizes empirical behavior over theoretical guarantees. Understanding the root causes of **EOB**, such as dataset artifacts, training dynamics, or modality interplay, requires further study.

While the proposed GroundAttack is a practical contribution, its evaluation is limited. We have not conducted human studies to assess the quality, plausibility, or diversity of the generated distractors, relying instead on manual inspection and EOB reduction metrics. The grounding experiments use CLIP similarity as a proxy for visual relevance, but CLIP itself has known biases. Moreover, we did not compare our approach with alternative distractor generation methods (e.g., adversarial (Li et al., 2020), contrastive (Cao et al., 2025; Qu et al., 2024; Chung et al., 2020), or perturbation-based techniques (Çavuşoğlu et al., 2024; Geva et al., 2022)), nor did we test whether models trained on GroundAttack-augmented data generalize better or improve robustness across tasks. We leave these directions for future work.

Despite these limitations, our work offers valuable insights and a practical diagnostic framework for identifying and addressing a previously underappreciated failure mode in VQA tasks. We hope this encourages the development of more robust and trustworthy benchmarks for evaluating multimodal reasoning in future vision-language systems.

**Conclusions**

In this paper, we uncover a previously overlooked limitation in multiple-choice Visual Question Answering (VQA) benchmarks, which we term the Easy Negative Bias (EOB). This bias allows VLMs to correctly answer questions without ever reading them. Our systematic evaluation across six diverse VQA benchmarks and four SOTA VLMs reveals that EOB is pervasive, affecting more than 80% of questions and fundamentally undermining the credibility of benchmark accuracy as a measure of true multimodal reasoning.

To analyze the roots of EOB, we leverage CLIP-based grounding experiments and show that correct answer choices consistently exhibit higher visual-text alignment scores than distractors, even without the question. This imbalance suggests that many benchmarks unintentionally favour answers that are visually obvious, reducing the need for genuine cross-modal reasoning.

To address this flaw, we introduce GroundAttack, a practical and scalable method for generating visually and semantically grounded adversarial negative options. By augmenting existing benchmarks with harder, more plausible distractors, GroundAttack significantly mitigates EOB, thereby restoring the need for the question and enhancing the robustness of VQA evaluation.

Our findings call for a rethinking of how VQA benchmarks are constructed and evaluated in the era of powerful pretrained VLMs. Future benchmarks must go beyond merely testing factual recall or language priors and ensure that answering truly depends on integrating both vision and language inputs, especially the question.

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
