## A GROUNDATTACK AND SELECTOR $\psi_s$ TYPES

To facilitate better understanding and implementation, we present a detailed pipeline of GroundAttack and various strategies for the selector $\psi_s$ in Figure 6. The pipeline below displays the illustrative frameworks introduced in Section 2.

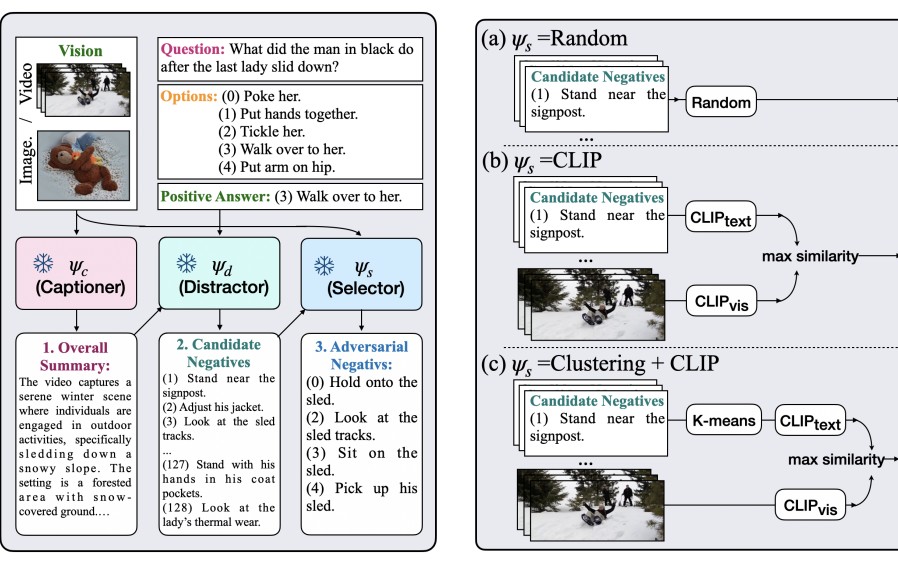

(a) GroundAttack          (b) Selector $\psi_s$

Figure 6: **GroundAttack** generates adversarial negative options that are more confusing, diverse, and visually groundable than original negatives. It mitigates Easy-Options Bias in VQA benchmarks through three components: (1) the Captioner ($\psi_c$), which converts visual content into detailed descriptions; (2) the Distractor ($\psi_d$), which produces plausible, groundable negative candidates; and (3) the Selector ($\psi_s$), which identifies the most adversarial negatives.

## B PROMPTS FOR CAPTIONER $\psi_c$ AND DISTRACTOR $\psi_d$

We define the roles of the captioner $\psi_c$ and distractor generator $\psi_d$ as follows:

- We utilize GLM-4.1V-9B as the captioner $\psi_c$ to convert video or image-based visual inputs $V$ into descriptive textual captions $T$.

- For the distractor $\psi_d$, we employ Gemma-3n-E48 to generate candidate negative answers $O_c$ conditioned on the question $Q$, the correct answer $A$, and the visual captions $T$.

- For image inputs, captions are generated based on salient objects, attributes, spatial relationships, and the overall scene.

- For video inputs, captions focus on objects, locations, atmosphere, and dynamic actions.

### B.1 PROMPTS FOR IMAGES

We present the three prompts used in GLM-4.1V-9B for **image captions** as follows:

**Prompt for generating fact from image**

You are an assistant that generates descriptive facts about an image.
### Instructions:
1. Input: You will be given an image.
2. Task: Based on the image, produce a concise descriptive caption in one or two sentences.
3. Output format: Return the result strictly as a Python JSON string, using the following structure:
{
"fact": "string"
}

4. Constraints:
- Only output the JSON string, no explanations or additional text.
- All keys and string values must be enclosed in double quotes ('"').
- Ensure the JSON is valid Python syntax.

**Prompt for detect object bounding boxes from image**

You are an assistant that generates detailed object information from an image.

### Instructions:
1. Input: You will be given an image.
2. Task: Detect at least 6 objects. For each object, specify:
- color
- size
- texture
- bounding box coordinates

The bounding box must be represented as normalized percentages of the image dimensions, in the format [x_min, y_min, x_max, y_max], where each value is between 0.0 and 1.0.

3. Output format: Return the result strictly as a Python JSON string, using the following structure:

{ "objects": ["string", "string", ...],
"object_details": {
"object1": { "color": "string", "size": "string", "texture": "string", "bounding_box": [x_min, y_min, x_max, y_max] },
"object2": { "color": "string", "size": "string", "texture": "string", "bounding_box": [x_min, y_min, x_max, y_max] } } }
4. Constraints:
- Bounding box values must be floats between 0.0 and 1.0, representing percentages of the image dimensions.
- Only output the JSON string, no explanations or additional text.
- All keys and string values must be enclosed in double quotes ('"').
- Ensure the JSON is valid Python syntax.

**Prompt for extract action, spatial relations from image**

> You are an assistant that extracts actions, scene context, and spatial relations from an image.
>
> ### Instructions:
> 1. Input: You will be given an image.
> 2. Task: Based on the image, identify:
> - actions: Any motions or interactions happening.
> - human_actions: Specific actions performed by humans.
> - spatial_relations: Relative positions between key objects (e.g., "cup on table").
> - scene: A single sentence summarizing the overall setting (e.g., "A cozy café interior at dusk").
>
> 3. Output format: Return the result strictly as a Python JSON string, using the following structure:
>
> { "actions": ["string", "string", ...], "spatial_relations": ["string", "string", ...], "scene": "string" }
> 4. Constraints:
> - Only output the JSON string, no explanations or additional text.
> - All keys and string values must be enclosed in double quotes ('"').
> - Ensure the JSON is valid Python syntax.

**Prompt for generating distractors for Image VQA**

We present the prompt used in Gemma-3n-E48 for generating 128 **candidate negative options** as follows.

> You are an expert at generating challenging negative distractors for image-based question answering. Given an image description, a question, and its correct answer, generate 128 clearly and definitively incorrect answer options.
>
> ### Guidelines:
> 1. **Grounded in the image**: Each distractor must reference actual events, objects, or details mentioned in the image description.
> 2. **Specifically Incorrect**: None of the distractors should correctly answer the given question.
> 3. **Deceptively Similar**: Distractors should resemble the correct answer in format, length, or type, making them plausible at first glance.
> 4. **No Hallucinations**: Do not introduce objects, actions, or details not present in the image description.
>
> ### Example:
> [Image Description]: A white dog is lying on a pet bed.
>
> [Question]: What does the white dog do after going to the cushion?
> [Correct Answer]: Smells the black dog
>
> [Negative Options] (JSON format):
> {
> "new_negatives": {
> "0": "Lies down on the pet bed.",
> "1": "Walks toward the black dog.",
> "2": "Explores the pet bed.",
> "3": "Watches the black dog."
> ...
> } } ### Output:
> - Provide exactly 128 numbered negative options.
> - The output must be valid JSON following the structure above.
> - Ensure the output is UTF-8 encoded.

### B.2 PROMPTS FOR VIDEOS

We present the three prompts used in GLM-4.1V-9B for **video captions** as follows:

**Prompt for generating fact from video**

You are an assistant that generates descriptive facts about a video.

### Instructions:
1. **Input**: You will be given a few video frames.
2. **Task**: Based on these frames, produce a concise descriptive caption in few sentences.
3. **Output Format**: Return the result strictly as a Python JSON string, using the following structure:

{
"fact": "string"
}

4. **Constraints**:
- Only output the JSON string; no explanations or additional text.
- All keys and string values must be enclosed in double quotes (").
- Ensure the JSON is valid Python syntax.

**Prompt for detect objects from video**

You are an assistant that generates detailed object information from a video.

### Instructions:
1. **Input**: You will be given a few video frames.
2. **Task**: Detect at least 6 objects. For each object, specify:
- color
- size
- texture
- spatial relations between objects

3. **Output Format**: Return the result strictly as a Python JSON string, using the following structure:

{ "objects": ["string", "string", ...],
"object_details":
"object1":  "color": "string", "size": "string", "texture": "string" ,
"object2":  "color": "string", "size": "string", "texture": "string"  ,
"spatial_relations": [
"object1 on top of object2",
"object3 next to object4", ... ]
4. **Constraints**:
- Only output the JSON string; no explanations or additional text.
- All keys and string values must be enclosed in double quotes (").
- Ensure the JSON is valid Python syntax.

**Prompt for extracting actions from video**

You are an assistant that extracts actions, scene context, and spatial relations from a video.

### Instructions:
1. **Input**: You will be given a few video frames.
2. **Task**: Based on these frames, identify:
- **actions**: Any motions or interactions happening.
- **human_actions**: Specific actions performed by humans.
- **spatial_relations**: Relative positions between key objects (e.g., "cup on table").
- **scene**: A single sentence summarizing the overall setting (e.g., "A cozy café interior at dusk").

3. **Output Format**: Return the result strictly as a Python JSON string, using the following structure:

```
{
"actions": ["string", "string", ...],
"spatial_relations": ["string", "string", ...],
"scene": "string"
}
```

4. **Constraints**:
- Only output the JSON string; no explanations or additional text.
- All keys and string values must be enclosed in double quotes (").
- Ensure the JSON is valid Python syntax.

**Prompt for generating distractors for Video VQA** We present the prompt used in Gemma-3n-E48 for generating 128 **candidate negative options** as follows.

You are an expert at generating challenging negative distractors for video-based question answering. Given a video description, a question, and its correct answer, generate 128 clearly and definitively incorrect answer options.

### Guidelines:
1. **Grounded in the Video**: Each distractor must reference actual events, objects, or details mentioned in the video description.
2. **Specifically Incorrect**: None of the distractors should correctly answer the given question.
3. **Deceptively Similar**: Distractors should resemble the correct answer in format, length, or type, making them plausible at first glance.
4. **No Hallucinations**: Do not introduce objects, actions, or details not present in the video description.

### Example:
[Video Description]: A white dog is lying on a pet bed.

[Question]: What does the white dog do after going to the cushion?
[Correct Answer]: Smells the black dog

[Negative Options] (JSON format):
```
{
"new_negatives": {
"0": "Lies down on the pet bed.",
"1": "Walks toward the black dog.",
"2": "Explores the pet bed.",
"3": "Watches the black dog."
...
} } ### Output:
```
- Provide exactly 128 numbered negative options.
- The output must be valid JSON following the structure above.
- Ensure the output is UTF-8 encoded.