# OpenReview forum: "GroundAttack: Mitigating Easy-Options Bias for Visual Question Answering"
_ICLR.cc/2026/Conference — ICLR 2026 Conference Withdrawn Submission_

### Official Review · Reviewer_ibiF · 2025-10-31

**Soundness:** 3
**Presentation:** 3
**Contribution:** 3
**Rating:** 6
**Confidence:** 3

**Summary:**

This paper highlights a caveat of certain multiple-choice Visual Question Answering (VQA) benchmarks used to asses the performance of Vision-Language Models (VLMs). Authors indeed present the “Easy-Options Bias” (EOB), where the correct option (ground-truth answer to the question) is much more aligned with the content of the input image than other options. In this case, considering the question is not necessary to answer it. Such an alignment is empirically validated by showing that the ground-truth option is often closer to the image in CLIP embedding space. To mitigate this limitation, they propose a method to generate hard negatives, i.e. alternative options to the ground-truth answer which are more aligned with the input image, to require any model to process the question to successfully answer it. Such method is composed of 3 modules: (i) a visual captioner generates a visual description of the image, fed to (ii) a distractor producing a set of negative candidate options that (iii) a selector will filter to keep the best ones. The existence of EOB, along with the impact their correction method, are validated empirically across different multiple-choice VQA benchmarks and VLMs.

**Strengths:**

- S1: High-quality benchmarks are primordial to properly and fairly assess the progress of a research field, so identifying and correcting potential flaws are important contributions.
- S2: The addressed limitation of VQA benchmarks is clearly explained, motivated and empirically validated.
- S3: The proposed method leads to a significant reduction of the detected bias in the considered VQA benchmarks.
- S4: It is also appreciated that authors present a comprehensive list of limitations for their work.

**Weaknesses:**

- W1: [Major] As acknowledged by authors, a comparison with other distractor generation methods would be important to evaluate the gain from the proposed method. From the current paper, we know that their method succeeds in making the VQA benchmarks more challenging, but maybe alternative methods could be even better.

- W2: [Major] Authors show that their method makes VQA benchmarks more challenging, but do they verify no hard negative is as good of an answer to the question as the ground-truth one, which would make the question ill-posed? Related to this question, are the generated options double-checked by human annotators?

- W3: [Major] Hard negatives are generated by models that could also be biased in a different way. Could authors elaborate on such limitations of using LLMs to automatically generate options?

- W4: [Major] It seems that the “CLIP-Selector” baseline is quite competitive, with a performance close to “GroundAttack” while being much simpler. Could authors discuss how each component of their method is really necessary?

**Questions:**

- Q1: [Related to W1] Could authors compare their method to other distractor generation approaches (e.g. the ones they present in their paper)?

- Q2: [Related to W2] Do authors verify no hard negative is as good of an answer to the question as the ground-truth one, which would make the question ill-posed? Related to this question, are the generated options double-checked by human annotators?

- Q3: [Related to W3] Could authors elaborate on such limitations of using LLMs to automatically generate options?

- Q4 [Related to W4] Could authors discuss how each component of their method is really necessary?

---

### Official Review · Reviewer_zzQa · 2025-10-31

**Soundness:** 2
**Presentation:** 2
**Contribution:** 2
**Rating:** 2
**Confidence:** 4

**Summary:**

This paper identifies and formalizes Easy-Options Bias (EOB) in multiple-choice VQA: models can often answer correctly using only vision and options, without the question. The authors propose GroundAttack, an agentic pipeline that replaces only the negative options with visually groundable, hard distractors, and re-annotate MMStar and NExT-QA accordingly. Empirically, accuracies drop markedly under (V,Q,O) and approach chance under (V,O), indicating reduced shortcutting and a more realistic evaluation.

**Strengths:**

The paper shows EOB is widespread across several VQA benchmarks and defines EOB, offering a principled lens on a subtle evaluation flaw.

**Weaknesses:**

1. Method is primarily evaluative/incomplete. The work focuses on test-time re-annotation; it does not study whether training on GroundAttack-augmented data improves model robustness or transfer—leaving the algorithmic impact on learning untested.

2. Limited dataset coverage. Despite diagnosing EOB broadly, the mitigation is validated only on MMStar and NExT-QA; generality to other listed benchmarks remains unverified.

3. Quality control of generated negatives. Beyond aggregate accuracy drops, the paper lacks human studies or fine-grained error audits to verify that generated negatives are semantically plausible yet definitively incorrect and not introducing new annotation noise.

Overall, this is a timely and important problem with a practical evaluation tool; however, the method feels incomplete without training-time studies and broader validation.

**Questions:**

See weakness.

---

### Official Review · Reviewer_te8u · 2025-11-03

**Soundness:** 2
**Presentation:** 3
**Contribution:** 2
**Rating:** 4
**Confidence:** 4

**Summary:**

This paper identifies and investigates a new form of dataset bias in multiple-choice Visual Question Answering (VQA) benchmarks, termed Easy-Options Bias (EOB). The authors demonstrate that state-of-the-art vision-language models (VLMs) can often predict the correct answer using only the vision and options inputs, without requiring the question. This undermines the validity of existing benchmarks as tests of multimodal reasoning. To mitigate this, the paper proposes GroundAttack, an automated method that generates visually plausible and semantically coherent hard negative options using a three-agent system: (1) a Captioner that produces image descriptions, (2) a Distractor that generates negative candidates, and (3) a Selector that chooses the most confusing distractors via CLIP similarity and clustering. Experiments demonstrate that GroundAttack substantially reduces EOB, lowering VLM accuracies under (V,O) settings to near-random performance and yielding more realistic evaluations.

**Strengths:**

1. The discovery of Easy-Options Bias is original and highly relevant. The study evaluates six benchmarks and multiple VLM families, providing convincing evidence that EOB is widespread and non-trivial.
2. GroundAttack is a pragmatic contribution, requiring minimal manual intervention while effectively generating high-quality hard negatives. Results show that EOB is mitigated and that model accuracies decrease to near-random levels when the question is omitted.

**Weaknesses:**

1. While the Easy-Options Bias (EOB) phenomenon indeed exists, the authors’ interpretation may be overly strong. When a model is given only the image and must select the answer without the question, it is natural and even intelligent for it to choose the option most visually aligned with the image. This behavior does not necessarily imply shallow reasoning. If the newly introduced confusing negative options were truly negative and the model accuracy dropped significantly (see 2), that could support the claim of shallow reasoning. However, even in that case, expecting model performance to approach random chance may not be a reasonable criterion.。
2. The paper uses VLM to generate confusing negative options, but these are not always semantically incorrect. For instance, in Figure 5, the original dataset’s negatives (“a basketball,” “a volleyball,” “a soccer ball”) are clearly wrong, while GroundAttack produces alternatives like “a sporting good,” “a ball for experienced players,” or “a clean ball,” which are not factually incorrect but rather vague or underspecified. In this example, the correct answer (“a tennis ball”) is indeed more appropriate than these generated alternatives, but the difference lies in degree rather than categorical correctness. However, the generated “hard negatives” may sometimes represent alternative but equally reasonable expressions of the correct answer—so subtle that even humans might struggle to distinguish them—making the correct answer non-unique or even ambiguous. This undermines the reliability of the paper’s claims regarding EOB reduction.
3. As acknowledged in the Limitations section, the models themselves contain inherent biases. Since multiple VLM are used to generate different components of the negative options, the consistency and reliability of these outputs are difficult to trust. The resulting negatives may reflect correlated model biases rather than genuine adversarial diversity.
4. The paper introduces three separate modules (Captioner, Distractor, Selector), but provides limited empirical or conceptual justification for this design choice. Why must these be distinct components? Why can’t a single model directly generate the final adversarial negatives? The necessity of the Captioner, as well as the two-stage “generate candidates then select” process, is not convincingly demonstrated. More ablation or efficiency analysis is needed to support this design.
5. As noted by the authors themselves, there is no comparison with alternative distractor generation methods, nor any experiment showing whether training on GroundAttack-augmented data improves generalization or robustness. Without such evaluations, the effectiveness of GroundAttack remains insufficiently validated and its broader utility uncertain.

**Questions:**

Please refer to the Weaknesses.

---

### Official Review · Reviewer_Eo8V · 2025-11-06

**Soundness:** 2
**Presentation:** 2
**Contribution:** 1
**Rating:** 2
**Confidence:** 4

**Summary:**

This work explores a previously unexplored failure mode in VQA benchmarks, where the models can answer a question with only the image and the option pair without needing the question. It then proposes a new method called GroundAttack in order to generate visually and semantically plausible hard negatives to remove the bias of the correct option towards the image in the feature space. Their experiment on NExT-QA and MMStar shows that the performance of VLMs drop close to random baseline when only image and option is given as input.

**Strengths:**

* The identification of Easy-Options Bias provides a interesting perspective of current VQA benchmarks.
* GroundAttack is modular, easy to implement, and agent-based, potentially generalizable to other multimodal tasks.

**Weaknesses:**

* Lemma 1 shows no proof, and the result is not very obvious.
* Tables 2 and 3: The paper reports number of smaller family of models (capped within 7B). It doesn’t show how the bigger parameter models perform in this setting.
* All experiments shows single run number. No variance reporting is done.
* The number of datasets studied is limited in scope.
* The goal of hard negatives is to make it hard for the model reach to the correct answer using any shortcuts. From Figure 5. Some options are not just hard but based on the question context are candidate correct answer. Figure5: Example 1 - The clean ball can be a correct answer as well. Example 2- The misty fields can be a correct answer as well. The way these distractors are generated and using embedding based selection, it seems some of the generated “hard negative options” are just rephrasing of the correct answer or plausible answers.
* For the crucial steps of the proposed methods, this work uses small family of models like Gemma-3n-E48. This can hinder the quality of the generated samples.
* The paper's initial diagnosis shows that the model performance is still quite high with only (V,O) pairs but it shows no analysis as to why.
* Definition 1 seems redundant.

**Questions:**

* Section 3.3: As per the claim of the paper, the options of a VQA triplet are not completely unbiased. The correct option is biased in the feature space towards the images, thereby leading the model to claim that as the correct answer instead of the negative ones. The paper shows no insights into how the Total EOB as per Table 1, correlated to the performance drop in Table 2 and Table 3. If the total EOB of MMStar is 13.6%, how is the average drop in performance across all the models only 9.7 % in Table 1?
* Since the hard negatives are selected using top-m embeddings, what is the comparative difference in similarity between the generated negative options and the original negative options with the correct answer?
* As there is no threshold given while selecting the negative options, what stops this method from selecting options with 0.99 similarity with the correct option?
* In B.1 Prompt for generating distractors for Image VQA: In the in-context example, the image description shows no relation to a black dog as mentioned in the question, correct options and the generated negative. Can you share any of the reasoning traces of generative negative options candidates?
* The claim that the VLM finds shortcut in answer a question is simply based on the accuracy metrics. Is it possible to show any analysis to back up this claim?
* Was there any human evaluation conducted to measure distractor plausibility?
* Can you clarify or empirically validate the probabilistic assumptions in Lemma 1?
* Can you verify that there is a consensus between a large model like GPT-4o and the model you used in this work, for generating the negative options?

---

### Note · Authors · 2025-11-25

**Comment:**

We thank reviewer for their effort in reviewing, and would like to revise our paper in the future.

**Withdrawal Confirmation:**

I have read and agree with the venue's withdrawal policy on behalf of myself and my co-authors.